# SELF-EVOLVING VISION-LANGUAGE MODELS FOR IMAGE QUALITY ASSESSMENT VIA VOTING AND RANKING

**Wen Wen**[1*]    **Tianwu Zhi**[2*]    **Kanglong Fan**[1]    **Yang Li**[2]    **Xinge Peng**[2]
**Yabin Zhang**[2†]    **Yiting Liao**[2]    **Junlin Li**[2]    **Li Zhang**[2]
[1] City University of Hong Kong    [2]ByteDance Inc.

## ABSTRACT

Improving vision-language models (VLMs) in the post-training stage typically relies on supervised fine-tuning or reinforcement learning, methods that necessitate costly, human-annotated data. While self-supervised techniques have proven effective for enhancing reasoning capabilities, their application to perceptual domains such as image quality assessment (IQA) remains largely unexplored. In this work, we introduce **EvoQuality**, a novel framework that enables a VLM to autonomously refine its quality perception capabilities without any ground-truth labels. EvoQuality adapts the principle of *self-consistency* to the ranking-based nature of IQA. It generates pseudo-labels by performing pairwise majority voting on the VLM's own outputs to establish a consensus on relative quality. These pseudo-rankings are then formulated into a fidelity reward that guides the model's iterative evolution through group relative policy optimization (GRPO). By iteratively leveraging its own predictions, EvoQuality progressively refines the VLM's perceptual capability. Extensive experiments show that EvoQuality boosts the base VLM's zero-shot performance by 31.8% on PLCC across diverse IQA benchmarks. Remarkably, despite being entirely self-supervised, EvoQuality achieves performance that is competitive with, or even surpasses, state-of-the-art supervised VLM-based IQA models, outperforming these models on 5 out of 7 IQA benchmarks. Furthermore, the framework demonstrates significant flexibility, allowing it to be stacked with pre-trained IQA models to bolster generalization on unseen datasets. Codes and checkpoints will be available at https://github.com/bytedance/EvoQuality.

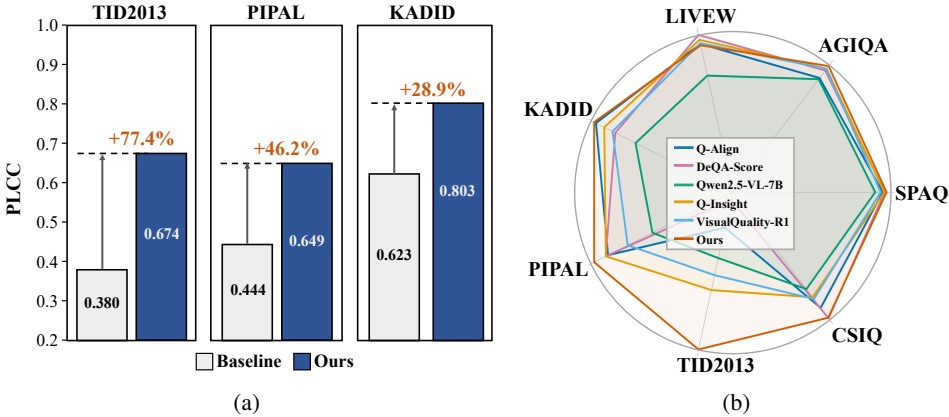

Figure 1: Without any ground truths, EvoQuality enables Qwen2.5-VL-7B to self-evolve its IQA capabilities, achieving (a) substantial performance improvements over the baseline and (b) superior or competitive results compared to supervised VLM-based models across multiple IQA benchmarks.

---

*Equal Contribution. Correspondence to Wen Wen (wwen29-c@my.cityu.edu.hk). † Project Lead.

# 1 INTRODUCTION

Vision-language models (VLMs) have demonstrated remarkable progress across a wide spectrum of tasks, from high-level vision applications such as image captioning and visual reasoning (Liu et al., 2023; Zhu et al., 2023; Li et al., 2023b; Wei et al., 2023; Wang et al., 2024; Bai et al., 2025) to low-level tasks including image quality assessment (IQA) (Wu et al., 2024a;c). Despite these advances, VLMs often exhibit instability and inconsistency, even for identical inputs, due to variations in model architectures, decoding strategies, and task complexities. To address these limitations, post-training has become a critical step in improving reliability and accuracy for downstream tasks. Existing approaches are predominantly based on supervised fine-tuning (SFT) and reinforcement learning (RL) with verifiable rewards. For instance, recent work has shown that online RL methods such as GRPO (Shao et al., 2024), combined with task-specific verifiable rewards (Guo et al., 2025; Zhou et al., 2025; Meng et al., 2025; Deng et al., 2025; Li et al., 2025b; Wu et al., 2025), can substantially enhance the reasoning capabilities of both large language models (LLMs) and VLMs.

Motivated by the increasing scale and capability of foundation models (Zhang et al., 2026), recent work has turned toward self-supervised post-training strategies that exploit the internal model outputs without reliance on ground-truth annotations. A prominent example is *self-consistency* (Wang et al., 2022), which leverages the "wisdom of the crowd" effect within the model itself. Instead of relying on a single prediction, self-consistency guides the model to generate multiple diverse reasoning traces for the same problem. The final answer is determined by aggregating these outputs, often through majority voting. The underlying intuition is that correct solutions are likely to emerge consistently across diverse reasoning paths, whereas incorrect ones are more sporadic. This paradigm has shown notable effectiveness: TTRL (Zuo et al., 2025) employs majority voting to improve mathematical reasoning in LLMs, MM-UPT (Wei et al., 2025) adapts it to VLMs for geometric problem-solving, and Fu et al. (2025) demonstrates entropy-based weighted voting to further balance accuracy and efficiency. Collectively, these advances highlight the potential of combining online RL with self-rewarding, enabling models to refine themselves without human supervision.

IQA aims to estimate perceptual image quality in alignment with human judgment. However, obtaining ground-truth labels is prohibitively costly, as it requires large-scale subjective studies. While some prior approaches (Gu et al., 2019) have employed off-the-shelf full-reference metrics (Wang et al., 2004; Zhang et al., 2014) to generate pseudo-labels, these methods are inherently restricted to synthetic datasets and their performance remains bounded by the capabilities of the selected metric. As a result, existing IQA datasets are limited in scale and diversity, constraining the generalization ability of supervised models and underscoring the need for self-supervised training paradigms. Nevertheless, extending self-evolving approaches to IQA poses unique challenges. Unlike discrete reasoning tasks, IQA is inherently continuous and subjective: there is no absolute correct score; instead, reliable evaluation relies on relative comparisons across images. This makes the design of effective self-supervision signals non-trivial. Recognizing this characteristic, recent IQA methods have incorporated learning-to-rank strategies, demonstrating strong performance (Zhang et al., 2021; Zhu et al., 2024; You et al., 2025; Wu et al., 2025). Therefore, adapting self-consistency to IQA requires addressing both how to aggregate votes and how to establish robust relative rankings.

In this paper, we present **EvoQuality**, a fully self-supervised framework that enables VLMs to progressively evolve their understanding of image quality via self-consistent voting and ranking. The framework operates in two stages. In the offline stage, the VLM performs pairwise comparisons and applies majority voting to produce high-confidence pseudo-labels that capture relative quality. In the online stage, these pseudo-rankings are transformed into fidelity rewards, which guide model updates through group relative policy optimization (GRPO). This iterative process can be repeated and allows the VLM to continuously improve its perceptual capabilities without reliance on human annotations. Extensive experiments show that EvoQuality significantly advances the base VLM's zero-shot capabilities, improving performance by 31.8% on PLCC across diverse IQA benchmarks covering authentic, synthetic, and AI-generated distortions. Crucially, without access to any ground-truth labels, EvoQuality achieves performance that surpasses state-of-the-art supervised VLM-based IQA models. By validating the efficacy of a self-consistency, voting and ranking paradigm, EvoQuality establishes a new and scalable approach for building robust IQA models, paving the way for reliable quality assessment in domains where human annotations are scarce or entirely unavailable.

## 2 RELATED WORK

### 2.1 SELF-EVOLVING METHODS

Large language models (LLMs) often rely on alignment techniques, such as reinforcement learning from human feedback (RLHF) (Ouyang et al., 2022), to improve their performance. However, the high cost and scalability limitations of human annotation have led to the development of self-evolving methods, which allow models to autonomously enhance their capabilities using unlabeled data. A central approach within this framework is to derive rewards from the model's own outputs. One notable method is based on consensus and self-consistency (Wang et al., 2022), where the model generates multiple candidate solutions for a given task, and the agreement between these solutions serves as a pseudo-reward. This approach has been successfully applied to enhance the reasoning abilities of LLMs in domains such as mathematics and code generation, with methods like TTRL (Zuo et al., 2025), MM-UPT (Wei et al., 2025), and SRT (Shafayat et al., 2025). Specifically, MM-UPT further strengthens a model's math and geometry abilities by generating synthetic geometry problems. Alternative approaches involve using a powerful model as an automated judge (Pang et al., 2024) or leveraging the model's internal states, such as confidence scores (Zhao et al., 2026; Li et al., 2025a) or the entropy of its output distribution (Zhang et al., 2025). Recently, DeepConf (Fu et al., 2025) has leveraged the entropy of tokens to enhance reasoning efficiency and improve performance at test time.

### 2.2 IMAGE QUALITY MODELS

**Regression-based models**. The conventional approach to NR-IQA has been regression, where the goal is to predict an absolute quality score for a given image. Early methods relied on hand-crafted features derived from natural scene statistics (Mittal et al., 2012a;b) or distortion-specific cues. The paradigm later shifted to deep learning, where end-to-end neural networks were trained to map image pixels directly to a quality score (Kang et al., 2014; Talebi & Milanfar, 2018; Ke et al., 2021; Yang et al., 2022). This regression framework has recently been extended to VLMs. For instance, methods like Q-Align (Wu et al., 2024b) treat IQA as an ordinal task, training the VLM to generate tokens representing distinct quality levels which are subsequently converted to numerical scores, while others like Q-Insight (Li et al., 2025b) pursue direct score regression. However, regression-based models often exhibit limited generalization and require complex perceptual scale realignment when trained on multiple datasets with different rating scales (Mikhailiuk et al., 2021).

**Ranking-based models**. To overcome the limitations of absolute scoring, ranking-based models reframe IQA as a relative task, positing that comparative judgments are more robust and consistent with human perception. This paradigm was pioneered by adapting learning-to-rank algorithms like RankNet (Burges et al., 2005), RRLRIQA (Gu et al., 2019), and psychometric models like the Thurstone model (Thurstone, 1927) to the IQA problem (Ma et al., 2017). By training on pairs of images, these models learn to predict which image has better quality. The fidelity loss, in particular, has been shown to be highly effective for this task (Tsai et al., 2007; Zhang et al., 2021). Recently, this ranking philosophy has been successfully applied to VLMs, leading to models such as Compare2Score (Zhu et al., 2024), DeQA-Score (You et al., 2025), and VisualQuality-R1 (Wu et al., 2025), which leverage the rich representations of VLMs for relative quality assessment. Our proposed method, EvoQuality, builds upon this powerful ranking paradigm. Moreover, it takes a significant step forward by eliminating the dependency on human-annotated labels, instead enabling the model to autonomously evolve its understanding of relative quality through a self-supervised voting and reward mechanism.

## 3 EVOLVING IMAGE QUALITY ASSESSMENT

This section details the design of EvoQuality, the self-evolving IQA framework, as shown in Figure 2. Inspired by self-consistency and the inherently "relative" nature of IQA, our method combines pairwise majority voting with ranking to provide a paradigm specifically tailored for IQA tasks. The core of EvoQuality is an iterative, two-step loop: 1) an offline stage where a VLM generates pseudo-ranking labels through pairwise voting, and 2) an online stage where the VLM's policy is updated

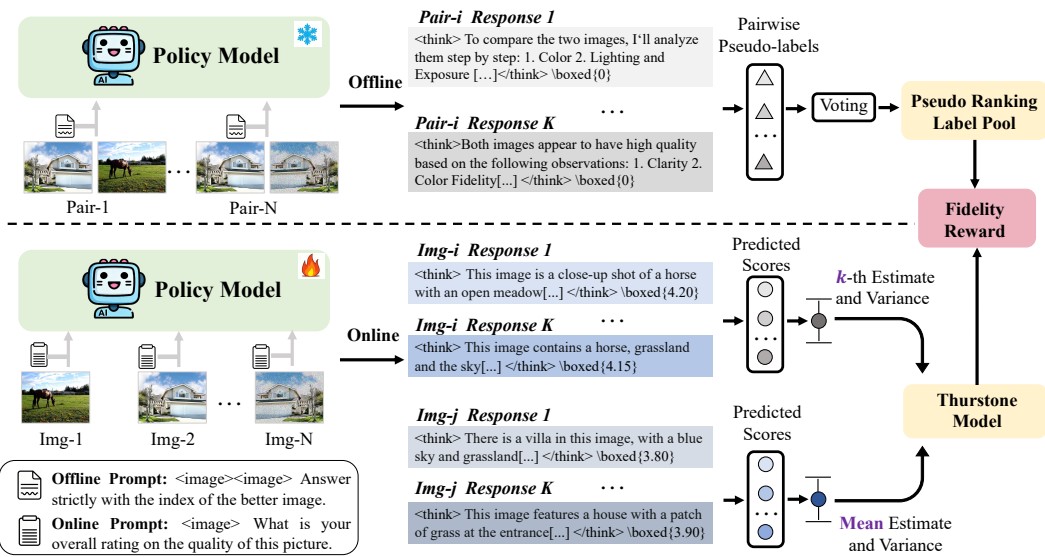

Figure 2: System diagram of the proposed self-evolving IQA framework EvoQuality. Each iteration operates in two stages. In the offline stage, the VLM generates pseudo-ranking labels for unlabeled image pairs via pairwise majority voting. In the subsequent online stage, the VLM's policy is updated via GRPO (Shao et al., 2024) using a fidelity reward derived from these pseudo-labels. This iterative loop enables the VLM to self-evolve its understanding of image quality.

using these self-generated labels as a reward signal. This loop allows the model to progressively refine its perceptual judgment.

## 3.1 OFFLINE PSEUDO-LABEL GENERATION VIA PAIRWISE MAJORITY VOTING

The offline stage in each evolution cycle is to generate a reliable supervisory signal from a batch of unlabeled images $\{x_1, x_2, \ldots, x_N\}$. Using the current VLM policy $\pi_\theta$ and a comparative prompt $c_{\text{compare}}$ (see Table 1), we perform pairwise comparisons on a set of random selected image pairs $\mathcal{P} = \{(x_i, x_j), \ldots\}$. To ensure self-consistency, we query the policy $\pi_\theta(\cdot | c_{\text{compare}}, x_i, x_j)$ $K$ times for each pair and determine the preference by majority voting. Besides, to mitigate positional bias in prompts, we randomly permute the order of image pairs.

Table 1: Structured prompt $c_{\text{compare}}$ for offline pairwise voting and prompt $c_{\text{score}}$ for online score generation, together with the reasoning suffix.

| |
|---|
| $c_{\text{compare}}$: `<image><image>` You are performing an image quality assessment task. Compare the two images and decide which one has better perceptual quality. Answer strictly with the index of the better image: $0$ if the first image is better, or $1$ if the second image is better. |
| $c_{\text{score}}$: `<image>` You are doing the image quality assessment task. Here is the question: What is your overall rating on the quality of this picture? The rating should be a float between $1$ and $5$, rounded to two decimal places, with $1$ representing very poor quality and $5$ representing excellent quality. |
| Suffix: You FIRST think about the reasoning process as an internal monologue and then provide the final answer. The reasoning process MUST BE enclosed within `<think> </think>` tags. The final answer MUST BE put in `boxed{}`. |

Unlike methods that aggregate votes into continuous quality scores, we directly record the outcome of each comparison. This process yields a set of high-confidence pairwise pseudo-labels, denoted as $p^*(x_i, x_j)$ for each pair $(x_i, x_j) \in \mathcal{P}$. The value is set to $p^*(x_i, x_j) = 1$ if image $x_i$ is determined

---

**Algorithm 1** EvoQuality

---

**Input:** Unlabeled image pairs $\mathcal{P} = \{(x_i, x_j), \dots\}$ from unlabeled images $\{x_1, x_2, \dots, x_N\}$, current VLM policy $\pi_\theta$, reference VLM policy $\pi_{\text{ref}}$, voting budget $K$, number of batches $M$, batch size $B$, maximum round $T$;

**Output:** Updated VLM policy $\pi_\theta^\dagger$;

1: **for all** $t \leftarrow 1$ to $T$ **do**
2:     **for all** image pair $(x_i, x_j)$ in $\mathcal{P}$ **do**                            ▷ Offline Stage
3:         Query $\pi_\theta(\cdot | c_{\text{compare}}, x_i, x_j)$ $K$ times
4:         Perform majority voting to generate pseudo preference $p^*(x_i, x_j)$ (Eq. 1)
5:     **end for**
6:     Sample $M$ batch inputs $\mathcal{B} \subseteq \mathcal{P}$, where $|\mathcal{B}_m| = B$                ▷ Online Stage
7:     **for all** $m \leftarrow 1$ to $M$ **do**
8:         **for all** image pairs $(x_i, x_j)$ in batch $\mathcal{B}_m$ **do**
9:             Query $\pi_\theta(\cdot | c_{\text{score}}, x_i)$ and $\pi_\theta(\cdot | c_{\text{score}}, x_j)$ $K$ times
10:            Compute the fidelity reward $r_k(x_i)$ with pseudo labels $\{p^*(x_i, x_j)\}_{j \in \mathcal{P}_i}$ (Eq. 3)
11:            Swap the position of $x_i$ and $x_j$ to compute the fidelity reward $r_k(x_j)$ (Eq. 3)
12:            Compute relative advantages $\{a_k(x_i)\}_{k=1}^K$ and $\{a_k(x_j)\}_{k=1}^K$ from the rewards (Eq. 4)
13:            Update policy $\pi_\theta$ using the objective $\ell(\theta, \pi_{\text{ref}})$ (Eq. 5)
14:         **end for**
15:     **end for**
16: **end for**

---

to be better, $p^*(x_i, x_j) = 0$ if $x_j$ is better, and $p^*(x_i, x_j) = 0.5$ for a tie:

$$p^*(x, y) = \begin{cases} 1 & \text{if } K_x > K_y \\ 0.5 & \text{if } K_x = K_y \\ 0 & \text{if } K_x < K_y \end{cases}, \tag{1}$$

where $K_x$ is the total count of votes for image $x$ being better, and $K_y$ is the total count of votes for image $y$ being better. This set of definitive pairwise outcomes, derived purely from the model's internal consensus, completely replaces the need for external human mean opinion score (MOS) labels and serves as the "ground truths" for the subsequent online training stage.

## 3.2 Online Policy Evolution with Fidelity Rewards

In the online stage, we fine-tune the VLM policy $\pi_\theta$ using the pseudo-labels generated during offline voting as the sole source of supervision. For each image $x_i$ from the offline comparison, we use a direct estimation prompt $c_{\text{score}}$ (Table 1). Using the policy from the previous iteration, $\pi_{\theta_{\text{old}}}(\cdot | c_{\text{score}}, x_i)$, we sample $K$ reasoning trajectories and corresponding quality scores $\{o_k, q_k(x_i)\}_{k=1}^K$. The vector of scores is denoted $q(x_i) = [q_1(x_i), \dots, q_K(x_i)]^\top$.

Motivated by Thurstone model (Thurstone, 1927) and VisualQuality-R1 (Wu et al., 2025), we compute the model's predicted comparative probability $p_k(x_i, x_j)$ for each of the pairs $(x_i, x_j) \in \mathcal{P}$ that were evaluated in the offline stage:

$$p_k(x_i, x_j) = \Phi\left(\frac{q_k(x_i) - \mu(q(x_j))}{\sqrt{\sigma^2(q(x_i)) + \sigma^2(q(x_j)) + \gamma}}\right), \tag{2}$$

where $\Phi(\cdot)$ is the standard Gaussian cumulative distribution function. $\mu(q(x_j))$ and $\sigma^2(q(x_j))$ represent the mean and variance of the quality predictions for $x_j$, respectively. $\gamma$ is a small positive constant to avoid any potential division by zero.

The core of our self-supervised learning is the reward function $r_k(x_i)$. We adapt the fidelity measure by replacing any ground-truth preference with our pseudo-label preference $p^*(x_i, x_j)$ from Eq. 1. The reward is calculated only over the set of images $\mathcal{P}_i = \{x_j | (x_i, x_j) \in \mathcal{P}\}$ that were paired with

Table 2: PLCC and SRCC results of EvoQuality, reported together with percentage improvements (%) over the baseline model Qwen2.5-VL-7B. WAVG. means the weighted average result, with the weighting proportional to the number of images in each dataset.

| Dataset | Qwen2.5-VL-7B | | EvoQuality@Round1 | | EvoQuality@Round2 | |
|---|---|---|---|---|---|---|
| | PLCC | SRCC | PLCC ($\triangle$%) | SRCC ($\triangle$%) | PLCC ($\triangle$%) | SRCC ($\triangle$%) |
| KONIQ | 0.761 | 0.703 | 0.840(+10.4%) | 0.794(+12.9%) | 0.835(+ 9.7%) | 0.791(+12.5%) |
| SPAQ | 0.853 | 0.843 | 0.902 (+ 5.7%) | 0.899 (+ 6.6%) | 0.903 (+ 5.9%) | 0.900 (+ 6.8%) |
| AGIQA | 0.766 | 0.681 | 0.839 (+ 9.5%) | 0.777(+14.1%) | 0.831 (+ 8.5%) | 0.771(+13.2%) |
| LIVEW | 0.718 | 0.700 | 0.847(+18.0%) | 0.814(+16.3%) | 0.847(+18.0%) | 0.813(+16.1%) |
| KADID | 0.623 | 0.587 | 0.784(+25.8%) | 0.782(+33.2%) | 0.803(+28.9%) | 0.807(+37.5%) |
| PIPAL | 0.444 | 0.388 | 0.613(+38.1%) | 0.545(+40.5%) | 0.649(+46.2%) | 0.583(+50.3%) |
| TID2013 | 0.380 | 0.363 | 0.624(+64.2%) | 0.587(+61.7%) | 0.674(+77.4%) | 0.611(+68.3%) |
| CSIQ | 0.717 | 0.656 | 0.827(+15.3%) | 0.789(+20.3%) | 0.865(+20.6%) | 0.839(+27.9%) |
| WAVG. | 0.615 | 0.570 | 0.751(+27.4%) | 0.709(+29.6%) | 0.770(+31.8%) | 0.726(+33.7%) |

$x_i$ during the offline voting stage:

$$r_k(x_i) = \frac{1}{|\mathcal{P}_i|} \sum_{j \in \mathcal{P}_i} \left( \sqrt{p^*(x_i, x_j) p_k(x_i, x_j)} + \sqrt{(1 - p^*(x_i, x_j))(1 - p_k(x_i, x_j))} \right). \quad (3)$$

This reward signal provides precise guidance for the policy to align with its own aggregated judgments. We collect $K$ fidelity rewards for $x_i$ into the vector $r(x_i) = [r_1(x_i), \ldots, r_K(x_i)]^\top$ and compute the relative advantage $a_k(x_i)$ by standardizing rewards within the group:

$$a_k(x_i) = \frac{r_k(x_i) - \mu(r(x_i))}{\sigma(r(x_i))}. \quad (4)$$

The policy update of $\pi_\theta(\cdot|c_{\text{score}}, x_i)$ follows the regularized GRPO:

$$\ell(\theta, \pi_{\text{ref}}) = -\frac{1}{BK} \sum_{i=1}^{B} \sum_{k=1}^{K} \left( \min\left( \frac{\pi_\theta(o_k|c, x_i)}{\pi_{\theta_{\text{old}}}(o_k|c, x_i)} a_k(x_i), \text{clip}\left( \frac{\pi_\theta(o_k|c, x_i)}{\pi_{\theta_{\text{old}}}(o_k|c, x_i)}, 1 - \epsilon, 1 + \epsilon \right) a_k(x_i) \right) \right.$$
$$\left. - \beta \, D_{\text{KL}} \left( \pi_\theta(o_k|c, x_i) \| \pi_{\text{ref}}(o_k|c, x_i) \right) \right), \quad (5)$$

where $B$ denotes the batch size during online training. The reference policy $\pi_{\text{ref}}(\cdot|c, x_i)$ is fixed, while $\pi_{\theta_{\text{old}}}(\cdot|c, x_i)$ is the policy from the previous iteration, used to sample $K$ reasoning trajectories $o = \{o_k\}_{k=1}^{K}$. The KL divergence term is approximated as:

$$D_{\text{KL}} \left( \pi_\theta(o_k|c, x_i) \| \pi_{\text{ref}}(o_k|c, x_i) \right) \approx \frac{\pi_{\text{ref}}(o_k|c, x_i)}{\pi_\theta(o_k|c, x_i)} - \log \frac{\pi_{\text{ref}}(o_k|c, x_i)}{\pi_\theta(o_k|c, x_i)} - 1, \quad (6)$$

which constrains $\pi_\theta(\cdot|c, x_i)$ from diverging excessively from $\pi_{\text{ref}}(\cdot|c, x_i)$. The clipping threshold $\epsilon$ prevents unstable policy updates, while the coefficient $\beta$ balances the reward-weighted likelihood and KL regularization terms.

The whole algorithm is shown in Algorithm 1. This two-stage loop allows the model to progressively strengthen its understanding of image quality by iteratively generating and learning from its own evolving consensus.

## 4 EXPERIMENT

### 4.1 EXPERIMENTAL SETUPS

**Datasets**. EvoQuality is trained exclusively on the KONIQ dataset (Hosu et al., 2020), augmented with synthetically generated distortions. Following (You et al., 2024), we create 10 distorted variants

Table 3: PLCC and SRCC results of NR-IQA models trained on KONIQ. The proposed EvoQuality and handcrafted ones DO NOT access the ground truths. The top-2 results are highlighted in **bold** and underline. WAVG. means the weighted average result across the out-of-distribution datasets.

| Method | KONIQ | SPAQ | AGIQA | LIVEW | KADID | PIPAL | TID2013 | CSIQ | WAVG. |
|---|---|---|---|---|---|---|---|---|---|
| PLCC | | | | | | | | | |
| *Handcrafted* | | | | | | | | | |
| NIQE | 0.533 | 0.679 | 0.560 | 0.493 | 0.499 | 0.372 | 0.548 | 0.751 | 0.520 |
| BRISQUE | 0.225 | 0.490 | 0.541 | 0.361 | 0.442 | 0.412 | 0.337 | 0.749 | 0.449 |
| *Discriminative Deep-Learning-based* | | | | | | | | | |
| NIMA | 0.896 | 0.838 | 0.715 | 0.814 | 0.701 | 0.536 | 0.418 | 0.854 | 0.641 |
| HyperIQA | 0.917 | 0.791 | 0.702 | 0.772 | 0.555 | 0.511 | 0.169 | 0.816 | 0.560 |
| DBCNN | 0.884 | 0.812 | 0.730 | 0.773 | 0.562 | 0.469 | 0.008 | 0.682 | 0.521 |
| CONTRIQUE | 0.798 | 0.651 | 0.679 | 0.713 | 0.590 | 0.416 | 0.055 | 0.518 | 0.476 |
| Re-IQA | 0.870 | 0.197 | 0.578 | 0.671 | 0.451 | 0.354 | 0.047 | 0.529 | 0.361 |
| MUSIQ | 0.924 | 0.868 | 0.722 | 0.789 | 0.668 | 0.567 | 0.257 | 0.860 | 0.621 |
| CLIP-IQA+ | 0.909 | 0.866 | 0.736 | 0.832 | 0.762 | 0.550 | 0.293 | **0.900** | 0.641 |
| MANIQA | 0.849 | 0.768 | 0.723 | 0.849 | 0.526 | 0.603 | 0.172 | 0.712 | 0.582 |
| *VLM-based* | | | | | | | | | |
| Q-Align | 0.941 | 0.886 | 0.772 | 0.853 | 0.795 | 0.600 | 0.282 | 0.814 | 0.663 |
| DeQA-Score | **0.953** | 0.895 | 0.809 | **0.891** | 0.711 | 0.608 | 0.192 | 0.868 | 0.652 |
| Qwen2.5-VL-7B | 0.703 | 0.853 | 0.766 | 0.718 | 0.623 | 0.444 | 0.380 | 0.717 | 0.598 |
| Q-Insight | 0.928 | 0.893 | 0.818 | 0.870 | 0.759 | 0.608 | 0.483 | 0.759 | 0.704 |
| VisualQuality-R1 | 0.882 | 0.877 | 0.817 | 0.856 | 0.723 | 0.531 | 0.435 | 0.768 | 0.667 |
| EvoQuality@Round1 | 0.840 | 0.902 | **0.839** | 0.847 | 0.784 | 0.613 | 0.624 | 0.827 | 0.740 |
| EvoQuality@Round2 | 0.835 | **0.903** | 0.831 | 0.847 | **0.803** | **0.649** | **0.674** | 0.865 | **0.762** |
| SRCC | | | | | | | | | |
| *Handcrafted* | | | | | | | | | |
| NIQE | 0.530 | 0.664 | 0.533 | 0.449 | 0.430 | 0.120 | 0.110 | 0.645 | 0.349 |
| BRISQUE | 0.226 | 0.406 | 0.497 | 0.313 | 0.374 | 0.279 | 0.115 | 0.551 | 0.333 |
| *Discriminative Deep-Learning-based* | | | | | | | | | |
| NIMA | 0.859 | 0.856 | 0.654 | 0.771 | 0.693 | 0.486 | 0.422 | 0.844 | 0.616 |
| HyperIQA | 0.906 | 0.788 | 0.640 | 0.749 | 0.554 | 0.461 | 0.055 | 0.756 | 0.510 |
| DBCNN | 0.875 | 0.806 | 0.641 | 0.755 | 0.551 | 0.408 | 0.042 | 0.653 | 0.490 |
| CONTRIQUE | 0.754 | 0.632 | 0.606 | 0.676 | 0.437 | 0.341 | 0.069 | 0.318 | 0.411 |
| Re-IQA | 0.838 | 0.174 | 0.525 | 0.624 | 0.303 | 0.281 | 0.096 | 0.244 | 0.301 |
| MUSIQ | 0.929 | 0.863 | 0.630 | 0.830 | 0.650 | 0.535 | 0.248 | 0.815 | 0.592 |
| CLIP-IQA+ | 0.895 | 0.864 | 0.685 | 0.805 | 0.756 | 0.499 | 0.309 | **0.890** | 0.618 |
| MANIQA | 0.834 | 0.758 | 0.636 | 0.832 | 0.488 | 0.551 | 0.116 | 0.675 | 0.534 |
| *VLM-based* | | | | | | | | | |
| Q-Align | 0.940 | 0.887 | 0.735 | 0.860 | 0.792 | 0.518 | 0.284 | 0.750 | 0.631 |
| DeQA-Score | **0.941** | 0.895 | 0.729 | **0.871** | 0.716 | 0.565 | 0.132 | 0.847 | 0.614 |
| Qwen2.5-VL-7B | 0.761 | 0.843 | 0.682 | 0.700 | 0.587 | 0.388 | 0.363 | 0.656 | 0.554 |
| Q-Insight | 0.909 | 0.891 | 0.760 | 0.839 | 0.762 | 0.534 | 0.468 | 0.714 | 0.667 |
| VisualQuality-R1 | 0.852 | 0.878 | 0.760 | 0.827 | 0.719 | 0.486 | 0.388 | 0.707 | 0.631 |
| EvoQuality@Round1 | 0.794 | 0.899 | **0.777** | 0.814 | 0.782 | 0.545 | 0.587 | 0.789 | 0.699 |
| EvoQuality@Round2 | 0.791 | **0.900** | 0.771 | 0.813 | **0.807** | **0.583** | **0.611** | 0.839 | **0.719** |

of each original image by randomly sampling 10 out of 35 distortion types and 5 severity levels. To enforce a fully self-supervised setting, all ground truth information is discarded, including both the quality scores from KONIQ and the labels of distortion types and severities. Thus, EvoQuality treats the training corpus as an unlabeled image collection for its evolutionary process.

For evaluation, we conduct zero-shot testing on seven benchmarks: 1) authentic distortions: SPAQ (Fang et al., 2020), LIVEW (Ghadiyaram & Bovik, 2015); 2) AI-generated distortions: AG-IQA (Li et al., 2023a); 3) synthetic distortions: CSIQ (Larson & Chandler, 2010), TID2013 (Ponomarenko et al., 2015), KADID (Lin et al., 2019), and PIPAL (Gu et al., 2020).

**Competing methods**. We compare the zero-shot generalization performance of EvoQuality against a comprehensive suite of NR-IQA models, including: 1) traditional handcrafted methods, NIQE (Mittal et al., 2012b) and BRISQUE (Mittal et al., 2012a); 2) discriminative deep-learning-based models: NIMA (Talebi & Milanfar, 2018), HyperIQA (Su et al., 2020), DBCNN (Zhang et al., 2018), CONTRIQUE (Madhusudana et al., 2022), Re-IQA (Saha et al., 2023), MUSIQ (Ke et al., 2021), CLIP-IQA+ (Wang et al., 2023), and MANIQA (Yang et al., 2022); 3) VLM-based models: Q-Align (Wu et al., 2024b), DeQA-Score (You et al., 2025), Q-Insight (Li et al., 2025b) and

Table 4: PLCC and SRCC results of VLM-based models co-trained on KONIQ, SPAQ, and KADID datasets. The top-2 results are highlighted in **bold** and underline. WAVG. means the weighted average result across the out-of-distribution datasets.

| Method | KONIQ | SPAQ | KADID | AGIQA | LIVEW | PIPAL | TID2013 | CSIQ | WAVG. |
|---|---|---|---|---|---|---|---|---|---|
| PLCC | *in-distribution* | | | *out-of-distribution* | | | | | |
| Q-Align | 0.945 | 0.933 | **0.977** | 0.788 | 0.887 | 0.603 | 0.684 | 0.924 | 0.714 |
| DeQA-Score | **0.957** | **0.938** | 0.967 | 0.808 | **0.900** | 0.597 | **0.806** | **0.940** | 0.748 |
| VisualQuality-R1 | 0.915 | 0.902 | 0.935 | 0.815 | 0.869 | 0.606 | 0.703 | 0.899 | 0.722 |
| EvoQuality | 0.795 | 0.899 | 0.796 | 0.822 | 0.818 | 0.645 | 0.678 | 0.887 | 0.727 |
| VQR1+EvoQuality | 0.915 | 0.916 | 0.938 | **0.830** | 0.874 | **0.669** | 0.772 | 0.926 | **0.768** |
| SRCC | *in-distribution* | | | *out-of-distribution* | | | | | |
| Q-Align | 0.938 | 0.931 | **0.974** | 0.733 | 0.883 | 0.577 | 0.676 | 0.910 | 0.688 |
| DeQA-Score | **0.944** | **0.934** | 0.965 | 0.745 | **0.887** | 0.552 | **0.752** | **0.914** | 0.700 |
| VisualQuality-R1 | 0.899 | 0.903 | 0.931 | 0.752 | 0.850 | 0.554 | 0.633 | 0.839 | 0.666 |
| EvoQuality | 0.731 | 0.897 | 0.785 | 0.754 | 0.785 | 0.579 | 0.667 | 0.848 | 0.678 |
| VQ-R1+EvoQuality | 0.899 | 0.917 | 0.936 | **0.779** | 0.851 | **0.610** | 0.715 | 0.885 | **0.716** |

VisualQuality-R1 (Wu et al., 2025), and the pre-trained Qwen2.5-VL-7B (Bai et al., 2025) which serves as our baseline.

**Implementation details**. We conduct EvoQuality in two iterations ($T = 2$): the first uses only authentic image pairs from KONIQ, while the second incorporates synthetically generated images, with pairs randomly sampled under the same reference. In the offline stage of each iteration, 20,000 image pairs are randomly constructed, and pseudo-labels are obtained by prompting Qwen2.5-VL-7B (Bai et al., 2025) with prompt $c_{\text{compare}}$ in Table 1 to generate 32 responses ($K = 32$), followed by majority voting. In the online stage, the entire VLM is fine-tuned with GRPO (Shao et al., 2024), again generating 32 responses with prompt $c_{\text{score}}$ in Table 1. Training employs AdamW (Loshchilov & Hutter, 2017) with an initial learning rate of $3 \times 10^{-7}$, a linear decay schedule, and balance coefficient $\beta = 0.05$. Experiments are conducted on eight NVIDIA A100 GPUs with a per-GPU batch size of four, requiring about twelve hours per epoch.

## 4.2 RESULTS

**Significant improvement over the baseline.** As shown in Table 2, EvoQuality delivers consistent and substantial improvements over the Qwen2.5-VL-7B baseline across all eight IQA benchmarks. After just a single evolutionary iteration (EvoQuality@Round1), the model achieves notable gains, with the weighted average PLCC and SRCC increasing by 27.4% and 29.6%, respectively. Moreover, the second iteration (EvoQuality@Round2) furthur improves the weighted average performance by 31.8% and 33.7%. These results confirm the effectiveness of the self-supervised voting and ranking in enhancing the perceptual judgment of the base VLM.

**Superiority over supervised IQA models.** A key strength of EvoQuality lies in its exceptional generalization across diverse datasets. Despite being trained exclusively on unlabeled images from the KONIQ dataset, the model demonstrates robust performance on seven unseen test sets, covering a wide spectrum of content and distortion types, as shown in Table 3.

First, on a weighted average across all benchmarks, our final model, EvoQuality@Round2, surpasses the leading supervised model, Q-Insight, by more than 0.05 on PLCC and SRCC. Second, this robustness is particularly evident on the challenging TID2013 dataset, where all learning-based methods struggle. EvoQuality achieves a PLCC score over 0.15 higher than the next-best performer NIQE. Third, EvoQuality shows notable gains on the AGIQA dataset, which contains AI-generated content. Unlike prior supervised models that fail to generalize to such novel content, our method achieves substantially superior performance, demonstrating its capacity to adapt to emerging image types. Furthermore, EvoQuality outperforms VisualQuality-R1, a method that leverages reinforcement learning with pairwise fidelity rewards from ground truths. As observed in Table 3, although EvoQuality does not match supervised VLM-based models on the in-domain KONIQ dataset, its superior cross-dataset performance indicates that the proposed self-evolving paradigm reduces overfitting and enhances generalization.

Table 5: PLCC performance comparison of EvoQuality and EvoEstimate across two evolutionary rounds. The best results are highlighted in **bold**. $\Delta$ indicates an improvement of at least $0.01$ in the second round over the first.

| Dataset | Round0 | EvoQuality | | | EvoEstimate | | |
|---|---|---|---|---|---|---|---|
| | | Round1 | Round2 | $\Delta$ | Round1 | Round2 | $\Delta$ |
| KONIQ | 0.761 | 0.840 | 0.835 | ✗ | **0.844** | 0.802 | ✗ |
| SPAQ | 0.853 | 0.902 | **0.903** | ✗ | 0.902 | 0.902 | ✗ |
| AGIQA | 0.766 | 0.839 | 0.831 | ✗ | **0.841** | **0.841** | ✗ |
| LIVEW | 0.718 | **0.847** | **0.847** | ✗ | 0.819 | 0.815 | ✗ |
| KADID | 0.623 | 0.784 | **0.803** | ✓ | 0.744 | 0.753 | ✗ |
| PIPAL | 0.444 | 0.613 | **0.649** | ✓ | 0.624 | 0.633 | ✗ |
| TID2013 | 0.380 | 0.624 | **0.674** | ✓ | 0.552 | 0.550 | ✗ |
| CSIQ | 0.717 | 0.827 | **0.865** | ✓ | 0.843 | 0.840 | ✗ |
| WAVG. | 0.615 | 0.751 | **0.770** | ✓ | 0.738 | 0.736 | ✗ |

**Progressive refinement with more iterations.** As shown in Table 2, the second iteration (Evo-Quality@Round2), which incorporates synthetic images, further boosts performance, raising the weighted average PLCC and SRCC improvements to $31.8\%$ and $33.7\%$ over the baseline. The largest second-round gains appear on datasets with synthetic distortions, such as TID2013 (PLCC rising from $+64.2\%$ to $+77.4\%$) and PIPAL (PLCC rising from $+38.1\%$ to $+46.2\%$). Crucially, although the performance gains on authentic datasets are marginal, the model maintains its previous standards and does not suffer from catastrophic forgetting. These results highlight that EvoQuality's evolutionary process can progressively refine the model's capacity to handle a wider spectrum of image distortions.

**Enhanced sensitivity to synthetic distortions.** EvoQuality demonstrates particularly strong performance on datasets with challenging synthetic distortions, outperforming other VLM-based models, as shown in Table 3. Since these ground truths are obtained under controlled double-stimulus settings, the results indicate that EvoQuality's self-evolution by pairwise voting and ranking captures a more robust and generalizable model of perceptual quality. This confirms the effectiveness of learning relative preferences from self-generated signals for handling complex image degradations.

**Augmenting pre-trained IQA models for unseen generalization.** To further validate the flexibility of our framework, we conducted experiments following the multi-dataset co-training protocols of prior works (Wu et al., 2024b; You et al., 2025), utilizing the KONIQ, SPAQ, and KADID datasets, as shown in Table 4. While EvoQuality naturally trails supervised methods on in-distribution performance due to the absence of ground-truth labels, its primary advantage lies in its additive capability. EvoQuality can be applied not only to base VLMs but also to pre-trained, supervised IQA models. As demonstrated in Table 4, when using the pre-trained VisualQuality-R1 as the base model, Evo-Quality further boosts out-of-distribution performance. This unique characteristic underscores the practical applicability of EvoQuality in real-world scenarios, where it can be employed to enhance the generalization of existing models to unseen domains.

## 4.3 ABLATIONS

**Ablation on self-evolving variants**. To validate our choice of a ranking-based framework, we introduce EvoEstimate, a variant of EvoQuality. Unlike the proposed framework, which learns from relative quality comparisons, EvoEstimate is designed to predict a direct, absolute quality score. EvoEstimate vote and generate the score directly in the offline stage using the prompt $c_{\text{score}}$. The multiple voting results are then average to calculate the pseudo estimations. In the online stage, EvoEstimate is trained with the pseudo labels using the rewards as Q-Insight (Li et al., 2025b), treated IQA as a direct regression task. All other settings remain the same with EvoQuality. The comparison results are shown in Table 5.

Although EvoEstimate is competitive on some authentic image datasets, EvoQuality demonstrates significantly stronger performance on benchmarks with synthetic distortion datasets. This suggests that learning from relative rankings provides a more robust signal for understanding complex image

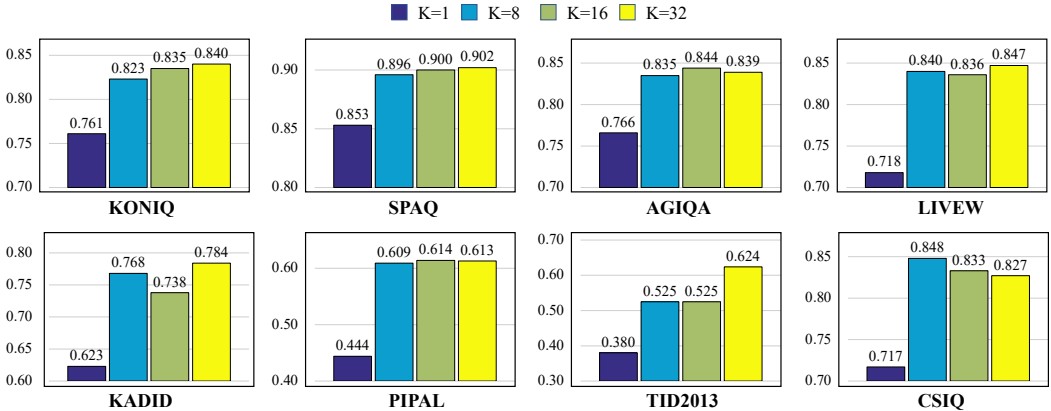

Figure 3: PLCC results of EvoQuality with varying $K$ candidate responses.

artifacts than regressing to an averaged pseudo-score. Furthermore, EvoQuality continues to improve in the second evolutionary round by $0.02$ on PLCC, whereas EvoEstimate shows almost no progress. These results indicate that pairwise voting and ranking provides a more reliable learning signal and enables improvements, while direct regression limits further evolution.

**Effect of candidate response** $K$. To analyze the effect of the number of candidate responses used for voting, we conducted an ablation study by varying $K \in \{1, 8, 16, 32\}$, while all other hyperparameters remained fixed. The case where $K = 1$ represents a baseline without any consensus mechanism. As depicted in Figure 3, the introduction of a voting process ($K > 1$) yields a significant performance improvement over the baseline. The results further indicate that a larger number of candidates generally leads to better outcomes, with $K = 32$ demonstrating the most stable and robust performance across the diverse IQA benchmarks.

## 5 CONCLUSION AND DISCUSSION

**Conclusion**. We have introduced EvoQuality, a fully self-supervised framework that enables a VLM to self-improve its IQA capability. By integrating pairwise majority voting with relative ranking strategies, EvoQuality generates robust supervisory signals, allowing the model to refine its perceptual judgments directly from internal consensus. Comprehensive experiments validate its effectiveness, demonstrating significant gains in the VLM's performance across diverse IQA benchmarks. Furthermore, EvoQuality demonstrates remarkable flexibility: it can be seamlessly applied atop existing supervised VLM-based models to augment generalization on unseen datasets, thereby establishing its practical value for real-world deployment.

**Limitations and future directions**. Despite the promising results, EvoQuality has limitations that suggest clear directions for future research. First, although the framework eliminates the reliance on ground-truth quality labels, it remains contingent upon an in-domain pool of unlabeled images. The diversity and scope of this unsupervised corpus may directly influence the VLM's ultimate performance. Future work could investigate methods to enhance data efficiency or enable effective learning from more generalized, out-of-domain image collections. Second, the self-evolution process, particularly the iterative voting and reinforcement learning stages, entails significant computational overhead. We posit that substantial optimizations are feasible. Future research could focus on developing more efficient voting schemes, such as active sampling strategies that prioritize the most informative image pairs. Finally, we envision extending the EvoQuality paradigm to other subjective perceptual tasks where relative ranking is intrinsically more suitable than absolute regression, such as assessing image aesthetics, evaluating the realism of generative models, or ranking user-generated videos. Collectively, these directions highlight the potential for developing robust, scalable, self-evolving perceptual systems.

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

## A  APPENDIX

## B  THE USE OF LARGE LANGUAGE MODELS (LLMS)

We utilized LLMs to assist in improving the grammar, clarity, and overall readability of this manuscript. All scientific claims, experimental results, and core ideas were conceived and articulated by the human authors. The authors have reviewed and edited all LLM-generated suggestions and take full responsibility for the final content of this paper.

