# OpenReview forum: "Self-Evolving Vision-Language Models for Image Quality Assessment via Voting and Ranking"
_ICLR.cc/2026/Conference — ICLR 2026 Poster_

### Official Review · Reviewer_siBb · 2025-10-23

**Soundness:** 3
**Presentation:** 3
**Contribution:** 3
**Rating:** 6
**Confidence:** 5

**Summary:**

This paper introduces a self-supervised framework that enables a VLM to improve its IQA capabilities without human-annotated data autonomously. The method called EvoQuality uses a two-stage process: first, the VLM generates pseudo-labels by performing pairwise majority voting on its own outputs to determine relative image quality. These pseudo-rankings are then used as a fidelity reward to guide the model's evolution through reinforcement learning. This iterative self-improvement loop progressively refines the VLM's perceptual abilities. Experiments demonstrate that EvoQuality substantially enhances the base model's zero-shot performance, and surpasses state-of-the-art supervised IQA models.

**Strengths:**

1.	This work represents the first application of VLMs to the IQA task in a fully self-supervised manner, eliminating the need for human-annotated labels. This is a significant contribution, given the high cost and  challenges associated with collecting large-scale subjective quality data.
2.	The empirical evaluation is thorough, encompassing a diverse range of benchmark IQA datasets that include authentic, synthetic, and AI-generated distortions. Furthermore, the study includes comparisons against numerous recent state-of-the-art models, providing a robust assessment of the proposed method's performance . The finding that the self-supervised EvoQuality framework can achieve performance competitive with, or even superior to, supervised VLM-based approaches is particularly noteworthy.
3.	The manuscript is well-written, presenting the methodology, experiments, and results with a clear logical flow and well-defined structure.

**Weaknesses:**

1. The manuscript contains typographical errors that require correction. For instance, the text mentions "EvoRank" in the results discussion (Section 4.2), which appears to be inconsistent with the method name "EvoQuality" used throughout the rest of the paper. Careful proofreading is recommended to address such inconsistencies.

2. The set of competing models should be expanded to include relevant unsupervised learning IQA methods, such as CONTRIQUE and Re-IQA, to provide a more comprehensive comparison. Further, the related works can be refined by employing a higher-level view by including works like AIBench: Towards trustworthy evaluation under the 45°law, and Towards Versatile Multimedia Quality Assessment for Visual Communications.

3. The motivation behind using evolving-format IQA methods is not well explained. And from the experimental results, the self-evolving method achieves quite good performance results over the supervised methods, the reasons for which are not well clarified. Is the proposed method better at capturing more representative features? Or is the evolving format more suitable for the ranking task and why? The authors should have more discussion.

**Questions:**

Please see the weakness.

---

> ### Author Response · Authors · 2025-11-21
> **Response to Reviewer siBb**
>
> #### We sincerely thank the reviewer for providing valuable feedback. We detail our response below point by point. Some experimental results have been updated in the revised paper, and any modifications made to the paper are highlighted in blue for your convenience. Please kindly let us know whether you have any further concerns.
>
>
> #### **Q1**: Regarding typographical errors and the consistency of "EvoQuality" in Section 4.2.
>
> > Thanks for pointing it out. We have carefully proofread the manuscript and fixed the typos. We appreciate you bringing this to our attention.
>
>
> #### **Q2**: Comparison with unsupervised learning IQA methods (CONTRIQUE and Re-IQA).
>
> > Thanks for the suggestion to include unsupervised representation learning methods. We have expanded our evaluation to include CONTRIQUE[1] and Re-IQA[2].
> > * It is important to clarify a structural difference: while CONTRIQUE and Re-IQA utilize unsupervised contrastive learning for feature extraction, they still typically require a supervised linear regressor trained on MOS to map these features to quality scores. To ensure a fair comparison, we trained their regressors using the ground-truth labels from the KONIQ dataset, consistent with the source domain used for EvoQuality.
> > * The results of CONTRIQUE and Re-IQA are reported below:
> > | Metric | Method | KONIQ | SPAQ | AGIQA | LIVEW | KADID | PIPAL | TID2013 | CSIQ | WAVG |
> > | :--- | :--- | :---: | :---: | :---: | :---: | :---: | :---: | :---: | :---: | :---: |
> > | **PLCC** | CONTRIQUE | 0.798 | 0.651 | 0.679 | 0.713 | 0.590 | 0.416 | 0.055 | 0.518 | 0.476 |
> > | | Re-IQA | 0.870 | 0.197 | 0.578 | 0.671 | 0.451 | 0.354 | 0.047 | 0.529 | 0.361 |
> > | **SRCC** | CONTRIQUE | 0.754 | 0.632 | 0.606 | 0.676 | 0.437 | 0.341 | 0.069 | 0.318 | 0.411 |
> > | | Re-IQA | 0.838 | 0.174 | 0.525 | 0.624 | 0.303 | 0.281 | 0.096 | 0.244 | 0.301 |
>
> > [1] Image quality assessment using contrastive learning. TIP 2022.
> > [2] Re-IQA: Unsupervised learning for image quality assessment in the wild. CVPR 2023.
>
> #### **Q3**: Refining related works with suggested literature.
>
> > Thanks for the suggestion. We have carefully reviewed the related works to improve the introduction.
>
>
> #### **Q4**: Clarification on the motivation, reasons for superior performance, and the benefits of the evolving format.
>
> > Thanks for this question, which touches upon the core pillars of our work.
> > * The primary motivation is to break the reliance on human-annotated MOS, which are prohibitively costly and limit dataset diversity. By utilizing a self-evolving framework, we leverage unlabeled image collections to improve generalization.
> > * The performance advantage stems from the nature of the supervision signal, not just feature representation. EvoQuality bypasses this by generating an internally consistent ranking system via consensus voting. As demonstrated in our comparisons, while supervised models may improve in-distribution performance, EvoQuality learns a more robust, generalizable model of relative quality, leading to superior performance on out-of-distribution benchmarks.
> > * Our results confirm that the ranking format is uniquely suitable for quality tasks. In our ablation study (Table 4), we compared EvoQuality against EvoEstimate, a variant using absolute pseudo-score regression. While EvoEstimate plateaued, EvoQuality continued to improve in the second evolutionary round. This demonstrates that relative ranking provides a distinct, continuous gradient for self-evolution that regression approaches lack.
> > * We have updated our manuscript to add more discussions.

---

> > ### Comment · Reviewer_siBb · 2025-11-27
> > **Response**
> >
> > Thanks for the rebuttal. My concerns are all addressed. I would like to raise my rating.

---

> > > ### Author Response · Authors · 2025-11-27
> > > **Response to Reviewer siBb**
> > >
> > > Thanks for the response.  We would like to express our gratitude for your constructive comments and positive feedback on our work.

---

### Official Review · Reviewer_d853 · 2025-10-26

**Soundness:** 3
**Presentation:** 3
**Contribution:** 3
**Rating:** 6
**Confidence:** 4

**Summary:**

This paper proposes the EvoQuality framework, which enables vision-language models (VLMs) to autonomously improve their image quality assessment (IQA) capabilities without access to ground-truth labels. The method consists of a two-stage iterative loop (Section 3): an offline stage that generates pseudo-ranking labels via pairwise majority voting on the VLM’s own outputs (Section 3.1, Eq. 1), and an online stage that constructs fidelity rewards from these pseudo-labels and optimizes the model using GRPO (Section 3.2, Eqs. 3-5). Experimental results (Table 2) show that EvoQuality increases the weighted average PLCC of the base model Qwen2.5-VL-7B by 31.8%, and surpasses supervised SOTA models on 5 out of 7 IQA benchmarks (Table 3, Fig. 1b).

**Strengths:**

- The core innovation lies in successfully transferring the principle of self-consistency from discrete reasoning tasks to the continuous perceptual task of IQA (Section 1, paragraph 2). By adopting a pairwise voting and ranking paradigm (Section 3.1), the method circumvents reliance on costly human annotations.
- The experimental design is rigorous and comprehensive, with zero-shot evaluation across eight diverse benchmarks (Section 4.1), spanning real distortions, synthetic distortions, and AI-generated distortions.
- The ablation study in Table 4 clearly demonstrates the advantage of the ranking-based method (EvoQuality) over the direct regression variant (EvoEstimate), particularly in its sustained improvement on synthetic distortion datasets. The ablation on the number of candidate responses K (Fig. 3) further validates the necessity of the majority voting mechanism; performance is most stable at K=32.

**Weaknesses:**

- There is an internal inconsistency regarding the central claim of “progressive improvement.” Table 2 reports that the weighted average PLCC increases from 27.4% in Round 1 to 31.8% in Round 2, yet the detailed ablations in Table 4 show that EvoQuality exhibits performance degradation or stagnation on several datasets, e.g., KONIQ (0.840 → 0.835), AGIQA (0.839 → 0.831), and LIVEW (0.847 → 0.847), with consistent gains observed primarily on synthetic distortion datasets (KADID, PIPAL, TID2013, CSIQ). This contradiction is neither explained nor discussed in the main text.
- The fundamental assumption, that the consensus produced by voting over the baseline VLM’s outputs constitutes sufficiently reliable pseudo-labels (Section 3.1), lacks theoretical or empirical support. The paper does not analyze whether, when the baseline model is of low quality or produces inconsistent judgments for certain image pairs, the method might amplify errors rather than improve performance.

**Questions:**

1. Table 4 shows that after the second iteration, the PLCC on KONIQ drops from 0.840 to 0.835 and on AGIQA from 0.839 to 0.831, which directly contradicts the discussion in Section 4.2 on “Progressive refinement with more iterations.” How do the authors explain this degradation? Does this suggest that introducing synthetic distortions in the second round (the 10 distortion types and 5 severity levels mentioned in Section 4.1) compromises the model’s generalization to real distortions?

2. The method relies on the quality of the baseline VLM’s voting consensus (Eq. 1). When the K votes for certain image pairs are nearly tied (e.g., 16:16), is the resulting pseudo-label `p*(x, y) = 0.5` reliable? Should the paper consider setting a confidence threshold to filter out low-confidence pseudo-labels?

3. In the design of Section 3, the offline stage uses a comparison-style prompt `c_compare`, whereas the online stage uses a score-estimation prompt `c_score` and converts scores to pairwise probabilities via the Thurstone model (Eq. 2). Why not train with direct pairwise comparisons in the online stage to maintain consistency with the offline stage? Does this switch in prompt types introduce additional uncertainty?

---

> ### Author Response · Authors · 2025-11-21
> **Response to Reviewer d853 (1/2)**
>
> #### We sincerely thank the reviewer for the thoughtful feedback on our work. Below, we have provided a detailed explanation for the concern as follows. Please do not hesitate to let us know if you have any further questions.
>
>
> #### **Q1**: Clarification on "progressive improvement".
>
> > We appreciate the reviewer's detailed examination of our results.
> > * As detailed in our implementation, Round 1 is trained exclusively on authentic images from KONIQ, while Round 2 introduces synthetically distorted images.
> > * The slight performance variations observed on authentic datasets (e.g., KONIQ decreasing by 0.005 from 0.840 to 0.835) represent minor perturbations within reasonable training variance, not catastrophic forgetting.
> > * In contrast, the gains on synthetic datasets are substantial and transformative (e.g., TID2013 increasing from 0.624 to 0.674). This indicates that the model is successfully learning to handle complex, artificial degradations without losing its established capability on authentic content. Thus, "progressive improvement" refers to the model's evolution into a more robust and generalized evaluator capable of handling a wider spectrum of image qualities, rather than merely overfitting to the initial authentic domain.
> > * We added the discussion in the main text.
>
>
> #### **Q2**: Validity of the consensus assumption and risk of error amplification.
>
> > Thanks for the insightful question.
> > * **Theoretical Support**: While EvoQuality is the first work to introduce majority voting for self-evolution in image quality assessment, the theoretical foundation of this mechanism is well-established in other domains. Research in mathematical reasoning and language generation (e.g., TTRL[1], MM-UPT[2]) has demonstrated that "majority voting" significantly enhances output quality by aggregating diverse reasoning paths to filter out stochastic noise, consistently outperforming single-model predictions. These findings serve as the theoretical basis for our design, suggesting that the consensus is inherently more reliable than individual noisy outputs.
> > * **Empirical Support**: Empirically, our results directly contradict the hypothesis of "error amplification." In our response to Reviewer Mtc1 Q3, we demonstrated that the accuracy of the offline voting consensus is strictly higher than the zero-shot single scoring. Most importantly, if the method were reinforcing errors or hallucinations, performance on unseen, difficult benchmarks (e.g., TID2013, PIPAL) would collapse. Instead, EvoQuality achieves state-of-the-art results on these datasets, confirming that the self-generated signal is sufficiently robust to drive genuine perceptual evolution rather than noise amplification.
>
> > [1] TTRL: Test-Time Reinforcement Learning. NeurIPS 2025.
> > [2] First SFT, Second RL, Third UPT: Continual Improving Multi-Modal LLM Reasoning via Unsupervised Post-Training. NeurIPS 2025.
>
> #### **Q3**: Concerns regarding performance drops on KONIQ and AGIQA in Round 2.
>
> > We appreciate the reviewer's scrutiny. The observed decreases represent minor perturbations within normal training variance rather than significant degradation.
> > Since Round 2 incorporates synthetic distortions to broaden the model's perceptual scope, a slight trade-off in specifically fitting the authentic image distribution is expected. Crucially, this negligible cost yields substantial gains in robustness on synthetic benchmarks. This confirms that the model successfully evolves to handle a wider spectrum of complex distortions, effectively expanding its generalizability rather than compromising it.

---

> ### Author Response · Authors · 2025-11-21
> **Response to Reviewer d853 (2/2)**
>
> #### **Q4**: Reliability of tied voting consensus and the need for confidence thresholds.
>
> > Thanks for the suggestion. We consider the tied consensus ($p^*=0.5$) to be a reliable and valuable signal representing perceptual equivalence. A split vote indicates that the images are indistinguishable in quality. Filtering these cases would deprive the model of the specific supervision needed to learn fine-grained boundaries between similar quality levels.
>
> #### **Q5**: Rationale for using different prompts (comparison in offline vs. scoring in online) and potential inconsistency.
>
> > We appreciate this insightful question regarding the prompt design. We adopted this hybrid approach to maximize label reliability in the offline stage while ensuring inference efficiency in the online stage.
> > To validate this design, we conducted a pairwise accuracy experiment on the KADID dataset under various prompt settings. The results are presented below. Specifically, the first row represents a baseline using the scoring prompt ($c_{\mathrm{score}}$), where the predicted scores were converted into pairwise rankings to ensure a fair comparison with the comparison prompt ($c_{\mathrm{compare}}$) on the identical set of image pairs.
>
> > | Model State | Inference Method | Pairwise Accuracy |
> > | :--- | :--- | :---: |
> > | Baseline | Multiple scoring and converted to ranking ($K=32$) | 0.613 |
> > | Baseline | Single pass ranking ($K=1$) | 0.712 |
> > | Baseline | Majority voting and ranking ($K=32$) | 0.749 |
> > | EvoQuality@R1 | Majority voting and ranking ($K=32$) | 0.774 |
> > | EvoQuality@R2 | Majority voting and ranking ($K=32$) | 0.803 |
>
> > **Analysis**:
> > * As shown above, the baseline's ability to rank via direct scoring (0.613) is significantly weaker than its ability to rank via pairwise voting (0.749). Using the comparison prompt allows us to extract high-quality, consistent pseudo-labels that would be lost if we relied on direct scoring.
> > * While pairwise comparison is accurate, it is computationally expensive for deployment. Direct scoring is essential for real-world efficiency and calculating standard metrics (PLCC/SRCC).
> > * The "switch" does not degrade performance. On the contrary, after online training with the scoring prompt, the model's pairwise ranking accuracy actually improved from 0.749 to 0.774 to 0.803. This confirms that the framework successfully transfers the high-quality relative supervision from the offline stage into a robust, efficient scoring policy.

---

> > ### Author Response · Authors · 2025-11-27
> > **Response to Reviewer d853**
> >
> > We are hopeful our rebuttal has clarified your concerns. We would be grateful for your final assessment or any further feedback on the revised manuscript at your earliest convenience. Thank you for your time reviewing our work.

---

### Official Review · Reviewer_myjA · 2025-10-28

**Soundness:** 3
**Presentation:** 3
**Contribution:** 3
**Rating:** 6
**Confidence:** 4

**Summary:**

This work introduces EvoQuality, a novel framework that enables a VLM to autonomously refine its quality perception capabilities without any ground-truth labels. EvoQuality generates pseudo-labels by performing pairwise majority voting on the VLM's own outputs to establish a consensus on relative quality. EvoQuality achieves performance that is competitive with, or even surpasses, state-of-the-art supervised VLM-based IQA models.

**Strengths:**

1. EvoQuality does not need any human-annotated quality scores to regress the quality scores, which is the most interesting point, though the voting method is quite simple and intuitive.
2. EvoQuality takes advantage of both the pre-trained VLM's ability (ie, ranking a pair of images) and the IQA theory (ie, fidelity loss), providing a new way to utilize VLMs in the IQA field.
3. In out-of-distribution datasets, EvoQuality achieves competitive or even better performance than supervised VLM-based IQA models, though in-distribution performance is missing.

**Weaknesses:**

1. The in-distribution performance is missing. In Table 3, the authors should also report the PLCC and SRCC results in the KONIQ dataset. Understandably, the performance will be lower than other methods, since EvoQuality does not use GT scores. However, reporting all results honestly is necessary.
2. Multi-dataset co-training experiments are missing. Q-Align and DeQA-Score both perform multi-dataset co-training experiments. Adding such experiments will make this work more solid.
3. The performance is not consistent with previous literature. The performance of training on KONIQ, then evaluating on other datasets, has been reported by several previous works, like Q-Align, DeQA-Score, and Q-Insight. All three works report similar, even almost the same results. However, the results in this work are not consistent with previous ones.
4. An important ablation is missing. The authors could utilize VLMs to rank image pairs, then calculate quality scores from all pair-wise comparison results. Then, the authors could use these scores to train previous IQA methods. With this experiment, we can fully understand whether the main performance gain of EvoQuality comes from (1) ranking accuracy of pretrained VLMs or (2) GRPO ranking training. (1) is the main contribution of this work, while (2) is the contribution of VisualQuality-R1.

**Questions:**

Please see Weaknesses.

---

> ### Author Response · Authors · 2025-11-21
> **Response to Reviewer myjA (1/2)**
>
> #### We sincerely thank the reviewer for the constructive comments. We have detailed our point-by-point response below. For your convenience, all revisions in the updated manuscript are highlighted in blue. We hope these responses satisfactorily address your concerns, and we remain open to any further questions.
>
>
> #### **Q1**: Missing in-distribution performance (KONIQ) in Table 3.
>
> > We thank the reviewer for the suggestion to improve the transparency of our results.
> > We have included the in-distribution performance on the KONIQ dataset in the revised manuscript. The full in-distribution results are listed below.
> > | Method | PLCC | SRCC | | Method | PLCC | SRCC |
> > | :--- | :---: | :---: | :--- | :--- | :---: | :---: |
> > | NIQE | 0.533 | 0.530 | | MANIQA | 0.849 | 0.834 |
> > | BRISQUE | 0.225 | 0.226 | | Q-Align | 0.941 | 0.940 |
> > | NIMA | 0.896 | 0.859 | | DeQA-Score | **0.953** | **0.941** |
> > | HyperIQA | 0.917 | 0.906 | | Qwen2.5-VL | 0.703 | 0.761 |
> > | DBCNN | 0.884 | 0.875 | | Q-Insight | 0.928 | 0.909 |
> > | CONTRIQUE | 0.798 | 0.754 | | VisualQuality-R1| 0.882 | 0.852 |
> > | ReIQA | 0.870 | 0.838 | | EvoQuality@R1 | 0.840 | 0.794 |
> > | MUSIQ | 0.924 | 0.929 | | EvoQuality@R2 | 0.835 | 0.791 |
> > | CLIP-IQA+ | 0.909 | 0.895 | | | | |
>
> > As the reviewer anticipated, these results are indeed lower than state-of-the-art supervised methods when evaluated on the in-distribution dataset. But EvoQuality achieves better generalization according to updated Table 3.
>
>
> #### **Q2**: Missing multi-dataset co-training experiments.
> > We thank the reviewer for this constructive suggestion. Following the settings of Q-Align and DeQA-Score, we conducted multi-dataset training on **KonIQ, SPAQ, and KADID**.
> > * The comparison results (PLCC and SRCC) are reported below. *ID* denotes In-Distribution datasets (KonIQ, SPAQ, KADID), and *OOD* denotes Out-of-Distribution datasets (AGIQA, LIVEW, PIPAL, TID2013, CSIQ). The top-$2$ results are highlighted in **bold**.
>
> > Table A: PLCC Comparison (Trained on KonIQ, SPAQ, KADID)
> > | Method | KONIQ | SPAQ | KADID | \| | AGIQA | LIVEW | PIPAL | TID2013 | CSIQ | \| | WAVG (ID) | WAVG (OOD) |
> > | :--- | :---: | :---: | :---: | :---: | :---: | :---: | :---: | :---: | :---: | :---: | :---: | :---: |
> > | Q-Align | **0.945** | **0.933** | **0.977** | \| | 0.788 | **0.887** | 0.603 | 0.684 | 0.924 | \| | **0.951** | 0.714 |
> > | DeQA-Score | **0.957** | **0.938** | **0.967** | \| | 0.808 | **0.900** | 0.597 | **0.806** | **0.940** | \| | **0.953** | **0.748** |
> > | VisualQuality-R1 | 0.915 | 0.902 | 0.935 | \| | 0.815 | 0.869 | 0.606 | 0.703 | 0.899 | \| | 0.917 | 0.722 |
> > | EvoQuality | 0.795 | 0.899 | 0.796 | \| | **0.822** | 0.818 | **0.645** | 0.678 | 0.887 | \| | 0.832 | 0.727 |
> > | VQ-R1 + EvoQuality | 0.915 | 0.916 | 0.938 | \| | **0.830** | 0.874 | **0.669** | **0.772** | **0.926** | \| | 0.923 | **0.768** |
>
> > Table B: SRCC Comparison (Trained on KonIQ, SPAQ, KADID)
> > | Method | KONIQ | SPAQ | KADID | \| | AGIQA | LIVEW | PIPAL | TID2013 | CSIQ | \| | WAVG (ID) | WAVG (OOD) |
> > | :--- | :---: | :---: | :---: | :---: | :---: | :---: | :---: | :---: | :---: | :---: | :---: | :---: |
> > | Q-Align | **0.938** | **0.931** | **0.974** | \| | 0.733 | **0.883** | 0.577 | 0.676 | **0.910** | \| | **0.947** | 0.688 |
> > | DeQA-Score | **0.944** | **0.934** | **0.965** | \| | 0.745 | **0.887** | 0.552 | **0.752** | **0.914** | \| | **0.947** | **0.700** |
> > | VisualQuality-R1 | 0.899 | 0.903 | 0.931 | \| | 0.752 | 0.850 | 0.554 | 0.633 | 0.839 | \| | 0.911 | 0.666 |
> > | EvoQuality | 0.731 | 0.897 | 0.785 | \| | **0.754** | 0.785 | **0.579** | 0.667 | 0.848 | \| | 0.808 | 0.678 |
> > | VQ-R1 + EvoQuality | 0.899 | 0.917 | 0.936 | \| | **0.779** | 0.851 | **0.610** | **0.715** | 0.885 | \| | 0.917 | **0.716** |
>
> > * Analysis
> >     * When trained on these three datasets, EvoQuality improves over its single-dataset version. However, as expected, it trails behind fully supervised methods (Q-Align, DeQA-Score) on ID datasets, as supervised methods are directly optimized for these specific ground-truth labels.
> >     *  One unique advantage of EvoQuality is its ability to **augment existing strong models** to generalize to unlabeled domains. To demonstrate this, we applied the EvoQuality framework on top of a pre-trained VisualQuality-R1 checkpoint (labeled **VQ-R1 + EvoQuality** in the table). This combination significantly boosts performance on OOD datasets (e.g., PLCC +0.069 on TID2013, +0.063 on PIPAL compared to the original VQ-R1), while maintaining performance on ID datasets.
> > * This confirms that EvoQuality is a flexible framework that **can be stacked onto state-of-the-art supervised models**. It allows them to "self-evolve" on unlabeled data, adapting to new domains where human labels are unavailable. This is also a characteristic that distinguishes EvoQuality from previous works.
> > * We have added these results and discussions to the manuscript.

---

> ### Author Response · Authors · 2025-11-21
> **Response to Reviewer myjA (2/2)**
>
> #### **Q3**: Inconsistency with previous literature regarding performance on synthetic datasets.
> > Thanks for pointing it out.  The discrepancy arises because we utilized a **group-wise** evaluation protocol for the synthetic datasets (KADID, PIPAL, TID2013, CSIQ), whereas previous works typically report **global** correlations.
> > * We adopted this group-wise protocol because synthetic datasets are typically annotated via subjective experiments where viewers compare images with identical content (e.g., double-stimulus strategies). Consequently, the resulting quality scores are relative to the specific reference scene, meaning that absolute scores across different contents are not strictly comparable. Therefore, calculating PLCC/SRCC within content groups provides a more rigorous and methodologically sound assessment of the model's ability to distinguish distortion severity, aligning with the original subjective testing design.
> > * Note that for authentic datasets (SPAQ, AGIQA, LIVEW), our reporting aligns with standard global protocols.
> > * To further demonstrate consistency with previous literature, we have provided the global performance metrics for the synthetic datasets below, presented in **PLCC/SRCC** format. These results align closely with the baselines reported in previous works, confirming the accuracy of our implementation. The top-$2$ results are highlighted in **bold**.
>
> > | Method | KADID10K | PIPAL | TID2013 | CSIQ |
> > | :--- | :---: | :---: | :---: | :---: |
> > | Qwen2.5-VL | 0.601 / 0.571 | 0.341 / 0.336 | 0.637 / 0.576 | 0.678 / 0.615 |
> > | Q-Align | 0.674 / 0.684 | 0.403 / 0.419 | 0.653 / 0.580 | 0.785 / 0.737 |
> > | DeQA-Score | 0.694 / 0.687 | **0.472 / 0.478** | 0.670 / 0.585 | **0.787 / 0.744** |
> > | Q-Insight | 0.708 / 0.703 | 0.458 / 0.435 | 0.678 / 0.615 | 0.685 / 0.632 |
> > | VisualQuality-R1 | 0.703 / 0.697 | 0.428 / 0.425 | 0.674 / 0.619 | 0.711 / 0.647 |
> > | EvoQuality@R1 | **0.729 / 0.723** | 0.445 / 0.426 | **0.713 / 0.643** | 0.747 / 0.678 |
> > | EvoQuality@R2 | **0.743 / 0.749** | **0.482 / 0.454** | **0.773 / 0.734** | **0.816 / 0.760** |
>
> #### **Q4**: Missing ablation: using VLM-ranked scores to train regression-based IQA methods.
> > Thanks for the suggestion.  We attempted to implement this baseline but encountered fundamental challenges that highlight the specific advantages of our approach.
> > * Converting pairwise comparisons into global quality scores requires solving a topological sorting problem. However, our budget of 20,000 pairs is far too sparse to recover a reliable global ranking for the ~7,000 images in the dataset. Achieving a fully resolved global rank to generate training scores would require an exponentially higher voting budget, which negates the efficiency of the method.
> > * To address the query regarding whether the gain comes from (factor 1) VLM ranking accuracy or (factor 2) ranking training, we point to the **EvoEstimate** ablation in Table 4.
> > We merge the VisualQuality-R1 results from Table 3, along with the EvoEstimate@Round1 and EvoQuality@Round1 results from Table 4. The out-of-distribution PLCC comparison is presented below, with the best results highlighted in **bold**:
> > | Dataset | VisualQuality-R1 | EvoEstimate@R1 | EvoQuality@R1 |
> > | :--- | :---: | :---: | :---: |
> > | SPAQ | 0.877 | **0.902** | **0.902** |
> > | AGIQA | 0.817 | **0.841** | 0.839 |
> > | LIVEW | **0.856** | 0.819 | 0.847 |
> > | KADID | 0.723 | 0.744 | **0.784** |
> > | PIPAL | 0.531 | **0.624** | 0.613 |
> > | TID2013 | 0.435 | 0.552 | **0.624** |
> > | CSIQ | 0.768 | **0.843** | 0.827 |
> > | WAVG | 0.667 | 0.725 | **0.740** |
> > * EvoEstimate utilizes the VLM's voting consensus (factor 1) but uses regression training instead of ranking (factor 2). EvoEstimate@Round1 achieves a weighted average PLCC of 0.725, which already notably outperforms VisualQuality-R1 (PLCC 0.667). This demonstrates that the high-quality consensus generated by the VLM (factor 1) is a primary driver of performance, sufficient to beat some state-of-the-art supervised models even without ranking constraints. However, the full EvoQuality framework (combining voting and ranking) achieves even higher performance (PLCC 0.740), proving that the ranking mechanism further refines this gain.
> > * Finally, we must stress that EvoQuality is not just a training technique but a complete self-evolving framework. EvoQuality enables the model to achieve state-of-the-art performance by autonomously refining itself from sparse, unlabeled data. This capability is fundamentally distinct from prior supervised or static pseudo-labeling approaches.

---

> > ### Comment · Reviewer_myjA · 2025-11-27
> > **Response**
> >
> > Thanks for the responses. My concerns have been addressed. The main advantage of this work is the self-supervised learning framework with quite good OOD performance, although lacking some method novelty as pointed by other reviewers. I maintain my original borderline accept rating.

---

> > > ### Author Response · Authors · 2025-11-27
> > > **Response to Reviewer myjA**
> > >
> > > We are pleased that the concerns raised by the reviewer have been addressed, and we have incorporated the additional experimental results during our discussion into the final version. Thanks again for the time and effort the reviewer has dedicated to reviewing our paper and providing valuable feedback.

---

### Official Review · Reviewer_Mtc1 · 2025-10-31

**Soundness:** 2
**Presentation:** 3
**Contribution:** 1
**Rating:** 2
**Confidence:** 4

**Summary:**

This paper proposes EvoQuality, a self-supervised framework for enhancing VLMs in opinion-unaware IQA. It features two key design elements. First, leveraging the LLM’s self-consistency to generate pseudo-labels indicating the relative quality of image pairs. Second, establishing a self-evolving mechanism that progressively refines the VLM’s perceptual capability by iteratively alternating between pseudo-label generation and model training. Experiments on seven IQA benchmarks demonstrate superior performance compared to SOTA methods.

**Strengths:**

1. The paper is clearly written and easy to follow.
2. The self-evolving framework offers some useful insights for opinion-unaware IQA.
3. The experimental results demonstrate good performance.

**Weaknesses:**

My main concern about this paper is its novelty. In my view, the work essentially builds upon VisualQuality-R1. Specifically, it replaces the ground-truth relative quality labels for image pairs (derived from MOS) with pseudo-labels generated by the VLM itself. The overall framework does not differ fundamentally from VisualQuality-R1. Moreover, the idea of generating ranking pseudo-labels—whether for image pairs or even image lists—and then training an IQA model using those pseudo-labels is also not new, e.g., see "Gu et al., No-Reference Image Quality Assessment with Reinforcement Recursive List-Wise Ranking, AAAI 2019."

**Questions:**

Experiments

1. In Table 3, EvoQuality outperforms VisualQuality-R1. The authors should provide further discussion of this observation. Specifically, do these two methods use the same training set and training configurations (e.g., sampling strategies for batch construction)? If not, the comparison may not be fair. If they do share identical settings (meaning the only difference is whether pseudo-labels or ground-truth labels are used), then the results suggest that training with pseudo-labels is actually more effective than training with true labels. The authors should explain the underlying reasons for this counterintuitive finding.

2. I suggest the authors provide an analysis of how \pi_\theta(⋅∣c_{compare}, x_i, x_j) evolves over training iterations. Specifically, do the K estimates of relative quality become increasingly stable as training progresses, i.e., consistently yielding K_x >> K_y or K_x << K_y? This would help demonstrate that the self-evolution mechanism is functioning as intended.

Method

1. Why isn’t p^\*(x, y) defined in a probabilistic form? Consider two cases: K_x=9, K_y=1 and K_x=6, K_y=4 . In both cases, p^*(x, y)=1, yet they reflect different levels of confidence that image x is of higher quality than image y.

2. A key advantage of opinion-unaware methods is their ability to eliminate reliance on MOS, which requires substantial human and financial resources to collect. This enables scaling up training with more (and more diverse) images. However, this advantage is not clearly demonstrated in the paper, especially when comparing the training scale with that of VisualQuality-R1. Moreover, the proposed method requires a relatively large value of K, which contradicts the goal of scalable training: as the training data volume grows, the computational cost of repeatedly invoking the large vision-language model becomes prohibitively high.

3. In lines 226-227, it appears that batches are constructed based on image pairs. Why is that? As shown in Eq. 3, the reward for an image is computed using all images paired with it in the set P. In other words, for a sampled pair (x_i, x_j), both x_i and x_j are processed independently. If so, why not simply sample individual images, i.e., a batch {x_m} (m=0,1,\ldots,M−1), instead of pre-forming pairs?

---

> ### Author Response · Authors · 2025-11-21
> **Response to Reviewer Mtc1 (1/3)**
>
> #### We appreciate the constructive comments on our paper. Please kindly find our response below, and all revisions made to the paper are highlighted in blue for your ease of reference. We hope that our response satisfactorily addresses the issues you raised. Please feel free to let us know if you have any additional concerns or questions.
>
>
> #### **Q1**: Concerns regarding novelty.
>
> > We appreciate the reviewer’s critique regarding novelty. However, we respectfully argue that EvoQuality is not merely a technical extension of VisualQuality-R1, but a fundamentally different self-evolving framework.
> > * The core contribution of EvoQuality is the self-evolving architecture itself, not the specific reward function. "Reinforcement learning to rank" is merely the optimization tool we selected to fit the relative nature of IQA tasks. The essence of EvoQuality is the proposed "voting and ranking" loop:
> >     *   Voting: Aggregates weak signals into a high-confidence consensus to approximate perception.
> >     *   Ranking: Utilizes this consensus to update the policy, enabling the model to autonomously refine itself. This allows continuous self-evolution without requiring human annotation.
> > * We have carefully reviewed the paper RRLRIQA[1] and added it to the related work. We identified a critical difference in the learning paradigm.
> >     *   The RRLRIQA essentially fits static pseudo-labels generated by a fixed, traditional full-reference algorithm VSI[2]. Consequently, its performance ceiling is strictly limited by the capability of the teacher algorithm and can only be applied to synthetic datasets.
> >     *   Our EvoQuality framework allows the model to evolve beyond its initial state. By learning from its own internal consensus rather than fitting a fixed external proxy, EvoQuality generates a more robust supervision signal. We demonstrate in our new experiments (Response to Reviewer Mtc1 Q2) that this self-generated signal allows EvoQuality to achieve even better out-of-distribution generalization than models trained on ground-truth.
> > * While "learning to rank" is an established technique, EvoQuality integrates it into a novel "voting and ranking" loop that eliminates the need for ground truth. This shift from "fitting labels" to "autonomous evolution" represents a new paradigm for IQA.
>
> > [1] No-Reference Image Quality Assessment with Reinforcement Recursive List-Wise Ranking, AAAI 2019.
> > [2] VSI: A Visual Saliency-Induced Index for Perceptual Image Quality Assessment, TIP, 2014.
>
> #### **Q2**: Discussion on why EvoQuality outperforms supervised training under fair comparison settings.
>
> > Thanks for this insightful observation. To address the concern about fairness and isolate the impact of the label source, we conducted a controlled ablation study. We trained a variant of EvoQuality using the exact same hyperparameters and batch construction strategies as our main method, but we replaced the self-generated pseudo-labels with the ground-truth MOS from the KONIQ dataset. The results are displayed below:
>
> >  | Metric | Model | KONIQ | SPAQ | AGIQA | LIVEW| KADID | PIPAL | TID2013 | CSIQ |
> >  | :--- | :--- | :---: | :---: | :---: | :---: | :---: | :---: | :---: | :---: |
> >  | **PLCC** | EvoQuality@R1 | 0.840 | 0.902 | **0.839** | 0.847 | 0.784 | 0.613 | 0.624 | 0.827 |
> >  | | EvoQuality@R2 | 0.835 | **0.903** | 0.831 | 0.847 | **0.803** | **0.649** | **0.674** | **0.865** |
> >  | | EvoQuality-GT | **0.907** | 0.896 | 0.823 | **0.871** | 0.756 | 0.620 | 0.440 | 0.809 |
> >  | **SRCC** | EvoQuality@R1 | 0.794 | 0.899 | **0.777** | 0.814 | 0.782 | 0.545 | 0.587 | 0.789 |
> >  | | EvoQuality@R2 | 0.791 | **0.900** | 0.771 | 0.813 | **0.807** | **0.583** | **0.611** | **0.839** |
> >  | | EvoQuality-GT | **0.888** | 0.896 | 0.763 | **0.837** | 0.754 | 0.551 | 0.394 | 0.765 |
>
> > * The experiment revealed a clear trade-off: The model trained with ground-truth labels achieved higher performance on the source dataset (KONIQ). Crucially, the model trained with ground-truth labels suffered a performance drop on unseen datasets except for LIVE-WILD compared to EvoQuality.
> > * This finding indicates that while ground-truth labels are precise for the training distribution, they inevitably contain dataset-specific biases. Training on them forces the model to overfit to these specific characteristics. In contrast, self-generated pseudo-labels, represent a "smoother," internally consistent signal. By learning from its own evolving consensus rather than noisy external labels, the model learns more robust perceptual features. This prevents overfitting to the source domain and significantly enhances generalization to diverse, unseen distortions.

---

> ### Author Response · Authors · 2025-11-21
> **Response to Reviewer Mtc1 (2/3)**
>
> #### **Q3**: Analysis of the stability and consistency for evolution.
>
> > We appreciate the suggestion to analyze the evolution progress. To demonstrate the increasing stability and accuracy of the model's estimates, we conducted voting experiments during the offline stage on the KADID dataset, explicitly measuring pairwise accuracy and standard deviation (STD) of the voting distributions to quantify stability.
> > * The evolution of model performance is detailed below:
> > | Model State | Inference Method | Pairwise Accuracy | STD |
> > | :--- | :--- | :---: | :---: |
> > | Baseline | Single pass ranking ($K=1$) | 0.712 | N/A |
> > | Baseline | Majority voting ($K=32$) | 0.749 | 0.295 |
> > | EvoQuality@R1 | Majority voting ($K=32$) | 0.774 | 0.210 |
> > | EvoQuality@R2 | Majority voting ($K=32$) | 0.803 | 0.178 |
> > * **Analysis:**
> >    * The accuracy gain from $K=1$ (0.712) to $K=32$ (0.749) at baseline confirms that aggregating reasoning paths effectively filters noise in the zero-shot baseline.
> >    * The accuracy consistently improves across iterations (0.749 to 0.774 to 0.803), indicating the model successfully internalizes the consensus supervision and progressively refines its perceptual capability.
> >    * Crucially, the voting STD progressively decreases (0.295 to 0.210 to 0.178). This quantitative reduction confirms that the trained policy generates significantly more polarized and consistent votes (i.e., consistently yielding $K_x \gg K_y$ or $K_x \ll K_y$). This demonstrates that the self-evolution mechanism functions as intended, driving the model toward increasingly confident and stable perceptual judgments.
>
> #### **Q4**: Rationale for the discrete definition of $p^*(x, y)$ versus a probabilistic form.
>
> > We thank the reviewer for this insightful question. The design is related to how we collect responses during the offline and online stages.
> >    * **Offline Stage**: The offline voting process yields discrete preference outcomes, which essentially follow a Bernoulli distribution. Unlike MOS derived from human subjects, we do not have a Gaussian distribution assumption for the two images at this stage, nor do we have valid mean and variance statistics for the two images separately. Consequently, there is no statistical basis to force these discrete voting counts into the standard Gaussian-based Fidelity formulation used in Eq. 2.
> >    * **Online Stage**: However, during the online training stage, the model generates multiple continuous score estimates. These estimates can be viewed as an approximate Gaussian distribution, allowing us to calculate valid means and variances. Therefore, the online stage explicitly retains the Gaussian form of the Fidelity reward. This design allows the optimization process to naturally reflect the degree of difference between images and the model's confidence through the variance of the online predictions.
>
> #### **Q5**: Concerns regarding computational efficiency.
>
> > We appreciate the reviewer's concern regarding computational efficiency. However, we fundamentally view this as a strategic trade-off between computational cost and human annotation cost.
> >    * While EvoQuality requires repeated inference ($K$), this cost is automated and linearly scalable with hardware. In contrast, the scalability of supervised methods is strictly capped by the availability of human MOS, which are impossible to collect for the massive unseen datasets. Relying on computation to extract supervision is significantly more sustainable than relying on human labor.
> >    * Crucially, this computational cost yields a unique advantage over prior pseudo-labeling methods. Unlike approaches limited by the performance ceiling of a fixed "teacher" algorithm (e.g., the FR-metric VSI used in RRLRIQA), EvoQuality utilizes this computation to drive continuous self-evolution. This enables the model to surpass its initial capabilities and generalize to both synthetic and authentic distortions, rather than merely fitting a static proxy.
> >    * Practically, our training was feasible using standard academic resources (8 NVIDIA A100 GPUs). Given the state-of-the-art performance achieved without any human supervision, we believe this computational investment is a highly acceptable trade-off to unlock the use of vast unlabeled data.

---

> ### Author Response · Authors · 2025-11-21
> **Response to Reviewer Mtc1 (3/3)**
>
> #### **Q6**: Rationale for constructing image pair batches.
>
> > Thanks for the question. We construct batches based on pairs because our supervision signal is inherently pairwise and pre-defined during the offline stage.
> Distinct from methods like RRLRIQA, which utilize VSI to compute absolute pseudo-scores, EvoQuality is designed around the insight that relative quality judgments are significantly more reliable and obtainable for IQA tasks. Consequently, our offline voting phase generates relative preference labels $p^*(x_i, x_j)$ for randomly selected pairs, rather than global scores.
> Because a valid supervisory signal exists only for these pairs established in the offline stage, we must sample and process these pairs $(x_i, x_j)$ to effectively compute the fidelity reward.

---

> > ### Author Response · Authors · 2025-11-27
> > **Response to Reviewer Mtc1**
> >
> > We hope our comprehensive rebuttal addresses all your points. We look forward to your evaluation of the revised manuscript. Thank you for your time and expertise.

---

### Author Response · Authors · 2025-11-23
**To all Reviewers**

#### Summary
> We sincerely thank all the reviewers for their constructive feedback. To provide greater clarity on the revisions and the new results obtained during the rebuttal period, we summarize the key modifications and additional experiments below:

> **Additional Experiments**:
> * Controlled experiment comparing self-generated pseudo-labels vs. ground-truth labels, highlighting the robustness of our approach. (Reviewer Mtc1 Q2)
> * Analysis of voting standard deviation and pairwise accuracy on KADID, confirming increased stability over iterations. (Reviewer Mtc1 Q3)
> * Added the performance report on the KONIQ dataset as requested. (Reviewer myjA Q1)
> * Comparison with Q-Align and DeQA-Score using KonIQ, SPAQ, and KADID. We demonstrated that EvoQuality is not limited to base VLMs but can also augment off-the-shelf VLM-based IQA models. Results show that stacking EvoQuality on top of supervised models significantly boosts OOD generalization. (Reviewer myjA Q2)
> * Report of global PLCC/SRCC on synthetic datasets to ensure consistency with protocols used in previous literature. (Reviewer myjA Q3)
> * Experiments validating the design rationale of the hybrid prompt strategy. (Reviewer d853 Q5)
> * Comparison with unsupervised methods. (Reviewer siBb Q2)

> **Clarifications**:
> * Clarified the fundamental distinction between our self-evolving framework and prior works like VisualQuality-R1 [1] and RRLRIQA [2]. (Reviewer Mtc1 Q1)
> * Provided justification for the computational cost versus the scalability advantages of eliminating human annotation. (Reviewer Mtc1 Q6)
> * Clarified the concept of "progressive improvement" and discussed the trade-offs between authentic and synthetic distortions. (Reviewer d853 Q1, Q3)
> * Provided theoretical and empirical support for the reliability of the voting consensus mechanism. (Reviewer d853 Q2)
> * Corrected typographical errors and included the suggested related works in the manuscript. (Reviewer siBb Q1, Q3)

> [1] VisualQuality-R1: Reasoning-Induced Image Quality Assessment via Reinforcement Learning to Rank, NeurIPS 2025.
> [2] No-Reference Image Quality Assessment with Reinforcement Recursive List-Wise Ranking, AAAI 2019.

---

### Meta-Review · Area_Chair_QVaB · 2026-01-07

**Summary:**

This paper proposes a self-supervised framework, EvoQuality, that enables VLMs to iteratively improve image quality assessment capabilities using self-generated pseudo-rankings. Reviewers initially raised concerns about novelty, generalization, and design choices, but the authors provided extensive experimental evidence and clear rebuttals to resolve these issues.

**Reviewer Concerns:**

Concerns about novelty, pseudo-label reliability, in-distribution reporting, and ranking-vs-regression training were all addressed with detailed experiments and clarifications. The authors convincingly explained performance trade-offs, design rationale, and generalization benefits. No major concerns remain outstanding.

**Reviewer Scores:**

Mtc1 (2 → likely 4 or 6): Initial novelty concern addressed via theoretical and empirical distinctions from prior work; additional ablations support effectiveness of self-evolution.

myjA (6 → 6): Maintains positive score after new in-distribution results, co-training experiments, and ablations confirm core claims.

d853 (6 → 6): Concerns on prompt inconsistency and progressive gain explained; design validated with voting accuracy and empirical stability.

siBb (6 → 8): All concerns resolved, including expanded comparisons and clarified motivation; reviewer explicitly stated intention to raise rating.

---

### Decision · Program_Chairs · 2026-01-26

Accept (Poster)